# Evaluation of the Anti-*Histoplasma capsulatum* Activity of Indole and Nitrofuran Derivatives and Their Pharmacological Safety in Three-Dimensional Cell Cultures

**DOI:** 10.3390/pharmaceutics14051043

**Published:** 2022-05-12

**Authors:** Carolina Orlando Vaso, Níura Madalena Bila, Fabiana Pandolfi, Daniela De Vita, Martina Bortolami, Jean Lucas Carvalho Bonatti, Rosângela Aparecida de Moraes Silva, Larissa Naiara Carvalho Gonçalves, Valeria Tudino, Roberta Costi, Roberto Di Santo, Maria José Soares Mendes-Giannini, Caroline Barcelos Costa-Orlandi, Luigi Scipione, Ana Marisa Fusco-Almeida

**Affiliations:** 1Department of Clinical Analysis, School of Pharmaceutical Science, Universidade Estadual Paulista, Araraquara 14800-903, SP, Brazil; carolovaso@hotmail.com (C.O.V.); niura.madalena.bila@gmail.com (N.M.B.); jeanlucasbonatti@gmail.com (J.L.C.B.); rosangela.moraes@unesp.br (R.A.d.M.S.); laarissagoncaalves@gmail.com (L.N.C.G.); maria.giannini@unesp.br (M.J.S.M.-G.); carolbarceloscosta@gmail.com (C.B.C.-O.); 2Department of Scienze di Base e Applicate per l’Ingegneria, Sapienza University of Rome, Via Castro Laurenziano 7, 00185 Rome, Italy; fabiana.pandolfi@uniroma1.it (F.P.); martina.bortolami@uniroma1.it (M.B.); 3Department of Environmental Biology, Sapienza University of Rome, Piazzale Aldo Moro 5, 00185 Rome, Italy; daniela.devita@uniroma1.it; 4Department of Chimica e Tecnologia del Farmaco, Sapienza University of Rome, Piazzale Aldo Moro 5, 00185 Rome, Italy; valeria.tudino@uniroma1.it; 5Department of Chemistry and Technology of Drug, Istituto Pasteur, Fondazione Cenci Bolognetti, Sapienza University of Rome, Piazzale Aldo Moro 5, 00185 Rome, Italy; roberta.costi@uniroma1.it (R.C.); roberto.disanto@uniroma1.it (R.D.S.)

**Keywords:** *Histoplasma capsulatum*, three-dimensional models, nitrofurans, indoles, pharmacological replacement, mechanisms of action, cytotoxicity, antifungal activity, drug repositioning, spheroids

## Abstract

*Histoplasma capsulatum* is a fungus that causes histoplasmosis. The increased evolution of microbial resistance and the adverse effects of current antifungals help new drugs to emerge. In this work, fifty-four nitrofurans and indoles were tested against the *H. capsulatum* EH-315 strain. Compounds with a minimum inhibitory concentration (MIC_90_) equal to or lower than 7.81 µg/mL were selected to evaluate their MIC_90_ on ATCC G217-B strain and their minimum fungicide concentration (MFC) on both strains. The quantification of membrane ergosterol, cell wall integrity, the production of reactive oxygen species, and the induction of death by necrosis–apoptosis was performed to investigate the mechanism of action of compounds **7**, **11**, and **32**. These compounds could reduce the extracted sterol and induce necrotic cell death, similarly to itraconazole. Moreover, **7** and **11** damaged the cell wall, causing flaws in the contour (**11**), or changing the size and shape of the fungal cell wall (**7**). Furthermore, **7** and **32** induced reactive oxygen species (ROS) formation higher than **11** and control. Finally, the cytotoxicity was measured in two models of cell culture, i.e., monolayers (cells are flat) and a three-dimensional (3D) model, where they present a spheroidal conformation. Cytotoxicity assays in the 3D model showed a lower toxicity in the compounds than those performed on cell monolayers. Overall, these results suggest that derivatives of nitrofurans and indoles are promising compounds for the treatment of histoplasmosis.

## 1. Introduction

Histoplasmosis is a systemic mycosis with worldwide prevalence caused by the dimorphic fungus *Histoplasma capsulatum*. This disease has been described in all continents except in Antarctica [1]. The worldwide annual incidence of histoplasmosis is around 500 thousand cases. In some outbreaks, 50% of infections result in clinical illness [2,3,4,5]. Humans acquire the infection through the inhalation of conidia or fragments of hyphae present in environment [4,6]. After inhalation, the fungus is phagocytosed by alveolar macrophages. Consequently, conidia change from the mycelial form to the yeast form, due to body temperature [6,7,8]

Histoplasmosis therapy varies according to the clinical condition and the immunological status of the patients [6]. The current antifungals used clinically for histoplasmosis belong to the classes of polyenes and azoles, among which amphotericin B (AmB) and itraconazole (ITZ) represent the gold standard [6,9,10]. The main drawbacks related to AmB is its high toxicity and its administration in a hospital setting [11,12]. This drug acts on ergosterol in the fungal membrane. Ergosterol is analogous to human cholesterol; therefore, this drug has high hepatic and renal toxicity, especially when used for long periods [10,11,12]. The azoles, on the other hand, can cause hepatotoxicity and problems related to drug interactions between different drugs that act on the cytochrome P450 system [13,14]. In addition, difficulties in penetrating the blood–brain barrier have been reported [4,14].

New therapeutic approaches have emerged over the years, such as nitrofuran and indole derivatives. Nitrofurans are described as antibiotics with a broad spectrum of action. They have been widely used in veterinary medicine for decades, and there are new drugs in this class approved for the treatment of Chagas disease [15,16,17,18]. However, due to problems related to toxicity, such as the appearance of mutations and cancers, as well as cases of pulmonary, cardiac, and reproductive system toxicity, hepatotoxicity, and nephrotoxicity, especially in long-term treatments, they have been banned in several countries [16,19,20]. In the scope of pharmacological replacement, additional activities of nitrofurans were discovered, such as antiparasitic and anti-proliferative agents [21,22,23]. Recently, some nitrofurans showed low toxicity when tested on human cell lines and potent antifungal and anti-biofilm activity against *Candida* species [24,25], with a capacity to inhibit cell adhesion and aggregation [26,27,28].

Indole sites represent extensively explored heterocyclic ring systems with different applications in pathophysiological conditions, such as cancer, inflammation, microbial, and viral infections [29,30,31,32]. Moreover, indole molecules have demonstrated activity against the planktonic and biofilm forms of *Candida albicans*, showing low toxicity, even at high concentrations, both in vitro and in vivo [25,33,34,35]. Furthermore, these compounds could inhibit *C. albicans* filamentation [35].

In light of previous discussions, we re-synthesized a series of nitrofuran (Figure 1) and indole (Figure 2) derivatives selected from our laboratory library [25,28,32], with the aim of identifying new anti-*Histoplasma* agents with low toxicity towards human cells.

In recent decades, cell culture has become one of the most used techniques to replace or complement the use of animals. However, due to the limitations that monolayer cultures (2D) present, three-dimensional (3D) cultures emerged [36]. These models have established an increasingly growing role, as their organizational structure refers to in vivo models, unlike cells that adhere and grow on a planar surface, as in the case of in vitro 2D culture [37,38].

Studies report that the 3D environment can better reproduce what happens in tissues, both physiologically and morphologically. Furthermore, there are differences in cellular responses, in the level of gene expression and cell behavior [39,40]. The scaffold-free technique, as in the case of spheroids produced in agarose gel, allows the cells to agglomerate and organize with each other [37,38]. This type of culture provides high interactions between cells and extracellular matrix. Despite a simple architecture, it can reproduce the in vivo environment, with high reproducibility in susceptibility assays [37,41,42]. Therefore, this study aims to verify the efficacy of nitrofuran and indole derivatives against *H. capsulatum*, as well as to evaluate their toxicity in the monolayer and in a scaffold-free 3D model (spheroids) of pulmonary cell lines and their possible mechanisms of action in the fungal cell.

## 2. Materials and Methods

### 2.1. Antifungal Drugs, Indole and Nitrofuran Derivatives

The antifungal drugs AmB and ITZ were purchased commercially (Sigma-Aldrich Milano, Italy).

The nitrofuran derivatives **1**–**17** [28], as well as the indole compounds **18**–**48**, **50**–**54** [25], and **49** [32], were synthesized according to literature, and all the analytical data were in accordance with those reported previously.

### 2.2. Microorganisms and Culture Conditions

Two strains of *H. capsulatum* (EH-315/BAC_1_ and G217-B/ATCC 26032) were used for all experimental tests. The EH-315 strain (BAC_1_) was registered in the World Federation for Culture Collection database under the number LIH-UNAM WDCM817 [43]. The G217-B strain (ATCC 26032) is classified as the American North strain (Nam2). The yeast *H. capsulatum* strains were maintained for 96 h at 37 °C on brain and heart infusion (BHI) agar (BD Difco™, Wokingham, Berkshire, UK) with 0.1% L-cysteine (Synth, Diadema, Sao Paulo, Brazil) and 1% glucose (Sigma-Aldrich Milano, Italy). The *H. capsulatum* was subcultured at 37 °C for 96 h and stirred at 150 rpm in Ham’s F-12 nutrient mixture (Ham-F12) (Gibco^®^, Thermo Fisher Scientific, Waltham, MA, USA) with 1.8% glucose (Synth, Diadema, Sao Paulo, Brazil), 0.1% glutamic acid (Synth, Diadema, Sao Paulo, Brazil), 0.6% buffer HEPES (Sigma-Aldrich Milano, Italy), and 0.008% L-cysteine (Synth, Diadema, Sao Paulo, Brazil) [44,45].

### 2.3. Susceptibility of H. capsulatum to Nitrofuran and Indole Derivates and Antifungal Drugs and Determination of Minimum Fungicide Concentration (MFC)

#### 2.3.1. Determination of Minimum Inhibitory Concentration (MIC_90_)

Susceptibility assays for *H. capsulatum* were performed, according to the M27-A3 document proposed by the Clinical Laboratory Standards Institute (CLSI) [46] with modifications proposed by Li and collaborators [47], Wheat and collaborators [48], and Kathuria and collaborators [49]. Briefly, stock concentrations of the nitrofuran and indole derivates were obtained by solubilizing them in 100% dimethylsulfoxide (DMSO) (Synth, Diadema, Sao Paulo, Brazil) at a concentration of 30,000 µg/mL and stored at −80 °C The antifungals AmB and ITZ were elaborated as recommendations in the M27-A3-CLSI document. The antifungals and compounds for solutions were solubilized in Roswell Park Memorial Institute (RPMI) 1640 medium (Gibco^®^, Thermo Fisher Scientific, Waltham, MA, USA), with the addition of 4-morpholinopropanesulfonic acid hemisodium salt (MOPS) (Sigma-Aldrich Milano, Italy) and glucose 2% (Synth, Diadema, Sao Paulo, Brazil). The concentration of the compounds ranged from 0.06 to 250 µg/mL, antifungals AmB (Sigma-Aldrich Milano, Italy) ranged from 0.007 to 4 µg/mL, and ITZ (Sigma-Aldrich Milano, Italy) ranged from 0.001 to 1 µg/mL. Initially, cell viability was verified using a Neubauer chamber (Kasvi-Sao Jose dos Pinhais, Parana, Brazil) and Trypan blue (Gibco^®^, Thermo Fisher Scientific, Waltham, MA, USA). Then, the inoculum was adjusted to a concentration of 1 × 10^6^ to 5 × 10^6^ cells/mL in 0.85% NaCl solution Then, a 1:10 dilution was performed, and the yeasts were placed in contact with serial dilutions of compounds and antifungals, with a final concentration of 5 × 10^4^ to 2.5 × 10^5^ cells/mL in RPMI-1640. The experiments were carried out in 96-well microplates (Kasvi, Sao Jose dos Pinhais, Parana, Brazil) with final a volume of 100 µL/well of the compounds and antifungals, following 100 µL of the inoculum. After incubation for 144 h at 37 °C at 150 rpm, visual reading was performed and 30 µL of 0.02% resazurin was added for colorimetric reading [50,51]. The initial screening of fifty-four compounds was performed on the EH-315 strain, and the most potent compounds (MIC_90_ ≤ 7.81 µg/mL) were tested on the ATCC G217-B strain. 

#### 2.3.2. Determination of Minimum Fungicide Concentration (MFC)

The MFC was determined, as described by Costa-Orlandi and collaborators [52], with a minor modification. In detail, 100 µL aliquots of the content of the wells were removed and transferred to Petri dishes containing supplemented BHI agar (BD Difco™, Wokingham, Berkshire, UK), and incubated at 37 °C for 96 h. Concentrations equal to or greater than the MIC_90_ were used. The MFC was defined as the lowest concentration of the compound that shows no growth of fungal colonies.

### 2.4. Cell Lines Mantainance

Two lung cell lines were used in cell culture assays: A549 (ATCC^®^ CCL-185) was characterized as an epithelial cell line and the MRC-5 cell line (ATCC^®^ CCL-171) was derived from fibroblasts. Cells were incubated in 5% CO_2_ at 37 °C in Dulbecco’s modified Eagle’s medium (DMEM) (Gibco^®^, Thermo Fisher Scientific, Waltham, MA, USA), adding 10% of fetal bovine serum (FBS) (Sigma-Aldrich Milano, Italy) [53]. To increase the cell–cell and cell–matrix interaction, the experiments were conducted, after at least two passages, post-thawing. The cells were only used for experimental assays of two passages after thawing, for greater cell–cell and cell–matrix interaction [37].

#### 2.4.1. Characterization of Three-Dimensional Cell Culture (3D) 

The 3D cellular spheroid model was developed as described by Friedrich and collaborators [37], with minor modifications. The 96-well plates (Kasvi, São José dos Pinhais, Paraná, Brazil) were coated with 50 µL of agarose (LCG) prepared at 1.5% in water. The cell lines were non-adhered in the bottom of the well, thus allowing the formation of a spheroid structure. Cells were prepared at concentrations ranging from 5 × 10^2^ to 1 × 10^4^ cells/well in DMEM medium supplemented with 10% fetal bovine serum (FBS) for both cell lines used. The suspensions were transferred to the agarose wells. Subsequently, the plates were incubated at 37 °C for 96 h in an atmosphere of 5% CO_2_ and humidified (standard condition) [37]. All experiments conducted for the spheroids were carried out for monolayer cultures, using the same concentration and incubation time.

##### Diameter Establishment

To evaluate the diameter of the spheroids over the days, images were captured at 96 h, 168 h, and 240 h (on the fourth, seventh, and tenth day of formation, respectively). For this, the spheroids were photographed with the In Cell Analyzer 2000 equipment (GE Healthcare, Chicago, IL, USA), located at the Clinical Mycology Laboratory at the School of Pharmaceutical Sciences, UNESP-Araraquara, Brazil. The diameter of the spheroids was verified using the Image J software.

##### Cell Quantification and Viability Using Trypan Blue

Eight spheroids from each cell line on the fourth, seventh, and tenth days of formation were transferred to conical tubes (TPP, Trasadingen, Switzerland), washed twice with 1× phosphate buffered saline (PBS) solution, and dissociated by the enzymatic action of trypsin (Gibco^®^, Thermo Fisher Scientific, Waltham, MA, USA). Dissociated cells were centrifuged, washed twice with 1× PBS, and stained with Trypan Blue (1:1 *v*/*v*) (Gibco^®^, Thermo Fisher Scientific, Waltham, MA, USA) using a hemocytometer. The same procedure was performed for monolayer culture wells.

##### Cell Viability—Qualitative

Metabolic activity was measured on the fourth, seventh, and tenth day, after adding 20 µL of the 60 µM resazurin solution (Sigma-Aldrich Milano, Italy) to the wells and incubating for 24 h in the standard condition. Absorbance was measured at 570 and 600 nm in a spectrophotometer (Epoch, Biotek, Santa Clara, CA, USA). Control wells contained only DMEM + 10% FBS used as blank, the cell monolayer (positive control), and wells with DMSO (negative control) [37,54]. 

##### External Cell Morphology of Spheroids

The topography of spheroids was visualized by scanning electron microscopy (SEM). The spheroids were transferred to 24-well plates, washed twice with 1× PBS, and fixed with 2.5% glutaraldehyde solution for 30 min. Then, the spheroids were washed twice with distilled water at room temperature (RT). Finally, the bottom of the plates was cut with a scalpel. The samples were mounted on silver aluminum cylinders and placed in a high vacuum evaporator for gold plating. The topographic characteristics of the spheroids were analyzed using a scanning electron microscope (Jeol JSM-6610LV at the Faculty of Dentistry, UNESP, Araraquara, Brazil) [55].

##### Confocal Fluorescence Microscopy

The spheroids were evaluated through fluorescence for their shape and size. They were transferred to 24-well plates so that each spheroid was in one well. The wells were washed twice with 500 µL of PBS buffer solution and 500 µL of paraformaldehyde solution 4% was added for fixed overnight. Then, the spheroids were washed with 500 µL of PBS, and 500 µL of Triton X 0.05% (Sigma-Aldrich Milano, Italy) was added for 30 min at RT. Afterwards, Triton X was removed, and 500 µL of bovine serum albumin (BSA) 2% (Sigma-Aldrich Milano, Italy) was added for 1 h at RT. As a marker, Alexa fluor 647 conjugated to phalloidin (Thermo Fisher Scientific, Waltham, MA, USA) was used, which binds to the actin of the cytoskeleton of the cells, emitting red fluorescence. The fluorescence marker was prepared in 2% BSA at a dilution of 1:2000, and 500 µL was added to each well containing the spheroids, which were then incubated for 1 h. Subsequently, Hoechst (Invitrogen—Thermo-Fisher-Scientific, Waltham, MA, USA), which stains nucleic acids blue, was prepared at a 1:2000 dilution in 2% BSA and added to the wells and incubated for 15 min. After labeling, the wells containing the spheroids were washed 2 times with 500 µL of PBS, and 500 µL of PBS was added. The images were obtained using the confocal fluorescence microscope (Carl Zeiss LSM 800 with Airyscan) of the Faculty of Dentistry of UNESP in Araraquara–SP, and the analyses were performed using the software Zen Blue 3.2 (Carl Zeiss, Jena, Germany) [55].

#### 2.4.2. Cytotoxicity Assay in Three-Dimensional (3D) and Monolayer (2D) Models by Resazurin Colorimetric Method

Cytotoxicity tests were performed for both strains (A549 and MRC-5), both in the three-dimensional model and in the monolayer, in order to verify the selectivity index (SI). After the incubation time, which was necessary for the formation of spheroids, 100 µL of the medium was removed from the plates of both models and 100 µL of different concentrations of the most potent compounds (0.48 to 250 µg/mL) was added, diluted in medium DMEM + 10% FBS and again incubated for another 72 h. Viability was verified by adding 50 µM of resazurin (Sigma-Aldrich, Milano, Italy), as described by Xião and collaborators [54]. Absorbance was measured in a spectrophotometer at 570 and 600 nm. The SI was calculated using the ratio between the CC_50_ and the MIC_90_ [56]. 

### 2.5. Determination of the Mechanisms of Action

#### 2.5.1. Ergosterol Dosage

Ergosterol quantification was performed according to the protocol described by Arthington-Skaggs and collaborators [57] with minor adaptations. Briefly, an inoculum was prepared at a concentration of 1 × 10^6^ cel/mL in HAM-F12 medium for strain EH-315 and placed together with sub-inhibitory concentrations of compounds and drugs. The compounds **11** (0.122 µg/mL), **32** (1.95 µg/mL), and **7** (0.06 µg/mL), representing a nitrofuran, an indole, and nitrofuran/indole derivatives, respectively, were used. The drugs AmB (0.03 µg/mL) and ITZ (0.06 µg/mL) were used as controls. The tubes were incubated under agitation at 150 rpm for 168 h. After incubation, tubes were centrifuged and cells were washed with 1 mL of sterile distilled water. Then, 3 mL of 25% KOH alcoholic solution was added to each tube and homogenized by vortexing for 1 min. Next, the suspensions were transferred to screwed glass tubes and incubated in a water bath at 85 °C for 1 h. One milliliter of sterile distilled water and three milliliters of n-heptane were added, after cooling the samples to RT. With the help of glass beads, the samples were shaken for 10 min to extract sterols. The n-heptane layer (the upper phase of the mixture) was transferred to microtubes and brought to −20 °C overnight. The analyses were carried out in a spectrophotometer at a wavelength of 281 nm with an ultraviolet (UV) spectrum [51,58,59,60,61]. Standard curves were prepared with 95% purity ergosterol (Sigma-Aldrich Milano, Italy) at concentrations ranging from 75 to 10 µg/mL.

#### 2.5.2. Evaluation of Morphology and Cell Wall Damage Using Calcofluor White Staining and Laser Scanning Confocal Microscopy

For cell wall morphology and damage, the dye calcofluor white was used, in 24-well plates (Kasvi, São José dos Pinhais, Paraná, Brazil) coated with glass coverslips. The inoculum and the compounds solutions were prepared according to Section 2.5.1 and dispensed onto 24-well plates containing round coverslips. The plates were incubated under agitation (150 rpm) for 144 h at 37 °C. After incubation, the plates were centrifuged, washed with 500 µL of PBS and 1 mL of 4% paraformaldehyde, and left overnight in the cold room (4 °C). The supernatant was removed and the solution of calcofluor white (Sigma-Aldrich Milano, Italy 100 mg/L) in PBS was placed, according to the manufacturer’s instructions, and incubated for 40 min at RT. The plate was centrifuged, washed with 500 µL of PBS, and the slide coverslip was mounted. The images were processed by the Software ZEN BLUE 2.3 System and the slides were observed using a confocal fluorescence scanning microscope (Carl Zeiss LSM 800 with Airyscan) at the Faculty of Dentistry, UNESP-Araraquara, Brazil [51].

#### 2.5.3. Quantification of Reactive Oxygen Species (ROS)

The intracellular production of ROS after treatment of *H. capsulatum* (EH-315) with the MIC_90_ concentrations of the compounds **7** (0.122 µg/mL), **11** (0.24 µg/mL), and **32** (3.90 µg/mL) was evaluated using 50 µM of H_2_DCFDA (2′,7′-dichlorodihydrofluorescein diacetate, Invitrogen- Thermo-Fisher-Scientific, Waltham, MA, USA). As a control, treatments with AmB (0.06 µg/mL), ITZ (0.125 µg/mL) and hydrogen peroxide (10 mM–H_2_O_2_) were used. The treatment was carried out as described in Section 2.5.1. After the incubation time (4 h), the samples were washed, and 500 µL of PBS was added, transferred to the cytometer tubes. Then, 1.5 µL of the H_2_DCFDA solution was added, incubated at RT, and protected from light for 10 min [62,63,64]. The samples were analyzed on a BD FACS Canto I flow cytometer localized at the Laboratory of Proteomics/Clinical Mycology, Faculty of Pharmaceutical Sciences, UNESP-Araraquara, Brazil.

#### 2.5.4. Apoptosis/Necrosis Assay

For the necrosis–apoptosis assay, the treatment was carried out as described in Section 2.5.3. The compounds **7**, **11**, and **32** were tested, and the drugs AmB and ITZ were used as controls. As markers, the apoptosis detection kit (Sigma-Aldrich Milano, Italy, A9210) was used following the manufacturer’s guidelines. The samples were incubated for 2 h at 37 °C protected from light. After the incubation time, the samples were centrifuged and washed with 1× PBS, and the cells were resuspended in 500 µL of kit buffer and later transferred to cytometer tubes. Analyses were accomplished on a BD FACS Canto I flow cytometer localized at the Laboratory of Proteomics/Clinical Mycology, Faculty of Pharmaceutical Sciences, UNESP-Araraquara, Brazil.

### 2.6. Statistical Analysis

For all tests, three technical and biological replicates were performed. To obtain the CC_50_, a non-linear semi-log regression was calculated. The generated results were submitted to statistical analysis using the *t* test or analysis of variance (one-way ANOVA) with Bonferroni post-test, using the GraphPad Prism 5.0 software (GraphPad Software Inc., La Jolla, CA, USA). Values of *p* < 0.05 were considered statistically significant.

## 3. Results

### 3.1. Determination of MIC_90_ and MFC

The results of the antifungal activity of all nitrofuran and indole derivatives against the EH-315 strain of *H. capsulatum* are shown in Table 1. The MIC_90_ values ranged from 0.122 µg/mL to greater than 125 µg/mL. 

The eighteen most potent compounds for the EH-315 strain (MIC_90_ ≤ 7.81 µg/mL) were tested for the ATCC G217-B strain (Table 2). MIC_90_ values for the ATCC G217-B strain ranged from 0.48 to greater than 250 µg/mL. In addition, the drugs AmB and ITZ were used as controls. The MIC_90_ values of AmB were 0.06 µg/mL (EH-315) and 0.03 µg/mL (ATCC G217-B), while the MIC_90_ values of ITZ were 0.125 µg/mL (EH-315) and 0.007 µg/mL (ATCC G217-B). All compounds showed a fungicidal profile in both strains, demonstrating MFC values equal to or double the MIC_90_ (Table 2).

### 3.2. Three-Dimensional Cell Culture

#### 3.2.1. Diameter

The concentrations of 3 × 10^3^ cells/well (for the A549 cell line) and 2 × 10^3^ cells/well (for the MRC-5 cell line) were established as ideal concentrations to produce spheroids, according to the previously established parameters. The images obtained and processed in ImageJ of the spheroids on the fourth day showed a mean diameter of 365.98 ± 66.95 µm for the A549 cell line and around 574.74 ± 52.03 µm for the MRC-5 cell line. On the seventh day, the mean diameter was 404.78 ± 49.92 µm for the A549 cell line and 460.38 ± 65.94 µm for the MRC-5 cell line; meanwhile, on the tenth day, the mean diameter was 441.80 ± 65.97 µm (A549) and 490.87 ± 32.23 µm (MRC-5), respectively (Figure 3). When the diameters were compared over the days, there was an increase in the diameter of the A549 strain on the tenth day compared to the fourth day (*** *p* < 0.001); meanwhile, for the MRC-5 strain, there was a reduction on the seventh day (*** *p* < 0.001) and the tenth day (*** *p* < 0.001) compared to the fourth day.

#### 3.2.2. Quantification and Cell Viability Using Trypan Blue

In both models and cell lines, there was an increase in cell proliferation compared to the day of plating. Overall, 3D culture had a lower cell proliferation rate than 2D culture. The models produced by the A549 cell line kept the cell quantity stable until the tenth day. The 2D model of the MRC-5 cell line showed an increase in cell proliferation on the seventh day (* *p* < 0.05). Regarding the 3D culture, there was a cellular increase on the seventh (*** *p* < 0.001) and on the tenth day (** *p* < 0.01), as shown in Table 3. 

Cell viability results from the trypan blue assay indicate a cell viability rate above 80%, even after ten days of culture for both cell lines used (A549 and MRC-5) and both models (2D and 3D) (Figure 4).

#### 3.2.3. Cell Viability Using the Resazurin Colorimetric Method

In the cell viability assay using the resazurin method, it was observed that the 3D models presented cell viability above 84%, even after ten days of formation, corroborating the results obtained with the trypan blue assay (Figure 5).

#### 3.2.4. Scanning Electron Microscopy

Figure 6 and Figure 7 show images obtained using the SEM technique on the spheroids of the A549 and MRC-5 cell lines on the fourth day of culture. Cells are organized in a spheroidal architecture, appearing smooth and rounded. The spheroids are intact, the cells interact with each other, and an extracellular matrix is formed, which is characteristic of this model.

#### 3.2.5. Confocal Fluorescence Microscopy

Figure 8 shows spheroids from both cell lines, A549 and MRC-5, on the fourth day of culture, labeled with Alexa fluor 647 conjugated to phalloidin and Hoechst. The spheroids are intact, compact, and have a regular spheroidal shape. The markings show a uniform distribution of cells within the spheroid. 

#### 3.2.6. Cytotoxicity Assay for Three-Dimensional and Monolayer Cell Culture

The compounds which showed a MIC_90_ equal or lower than 15.6 µg/mL on both strains were tested for their ability to produce toxicity in the monolayer and three-dimensional cell cultures. The CC_50_ was calculated for both the A549 and MRC-5 lineage models. In general, the compounds showed lower toxicity in 3D cultures compared to monolayers (Table 4 and Table 5). Most of the compounds showed greater selectivity for the fungal cells than for the host cells, with a range of SI from 2049.18 to 0.98. Among the tested compounds, **11** (nitrofuran derivative), **32** (indole derivative), and **7** (nitrofuran–indole derivative) were the most active and selective for H. capsulatum. Therefore, these compounds were chosen for further experiments which aimed to investigate their mechanisms of action. 

### 3.3. Determination of the Mechanisms of Action

#### 3.3.1. Ergosterol Dosage

The results of the ergosterol dosage indicate that the three compounds tested could significantly reduce the extracted ergosterol (** *p* < 0.01; *** *p* < 0.001), compared to the control. The drugs AmB and ITZ also significantly reduced the amount of ergosterol extracted, as expected (*** *p* < 0.001) (Figure 9). These results suggest that the compounds may inhibit ergosterol biosynthesis and/or bind to ergosterol present in the membrane.

#### 3.3.2. Evaluation of Morphology and Wall Damage

To check for possible wall damage, the fluorophore calcofluor white was used, which stains chitin. In the images in Figure 10, it was possible to verify that both the compounds **11** and **7**, in addition to the drugs AmB and ITZ, could deform the cells. Compound **11** caused small flaws in the contour of the fungal cell wall (Figure 10D). The cells treated with compound **7**, on the other hand, showed an increase in size and irregularity in their shape (Figure 10E). Only the cells treated with compound **32** had no structural change and/or damage (Figure 10F).

#### 3.3.3. Reactive Oxygen Species (ROS)

In the quantification of ROS (Figure 11), the indole derivative **32** and the nitrofuran/indole derivative **7** could generate a higher amount of ROS than the control (* *p* < 0.05). The AmB drug could generate a statistically significant amount of ROS compared to the control (*** *p* < 0.001). In addition, hydrogen peroxide, known for its production of ROS, also generated a greater amount compared to the control (*** *p* < 0.001).

#### 3.3.4. Cell Death Assay (Necrosis–Apoptosis)

All compounds (**7**, **11**, and **32**) and the drug ITZ (* *p* < 0.05; ** *p* < 0.01) induced necrotic cell death, with no significant induction of death by apoptosis. Differently, the fungal cells treated with AmB (** *p* < 0.01) had a cell death by apoptosis (Figure 12). 

## 4. Discussion

Although effective, therapeutic options for treating histoplasmosis are limited and demonstrate high risks of toxicity for the patient. Therefore, it is essential to discover new molecules with potent inhibitory properties and low toxicity [47,48,49,65,66,67,68]. In clinical practice, histoplasmosis requires long-term treatment, which can last up to 24 months, particularly for immunosuppressed individuals [69,70].

In the present work, fifty-four nitrofuran and indole derivatives were studied for their antifungal activity against *H. capsulatum*. Among these compounds, seventeen belong to the nitrofuran class (**1–6**, **8–17**), thirty-five are indole derivatives (**18–30**, **32–54**), while two molecules have both the nitrofuran and the indole moieties (**7** and **31**). Firstly, all the compounds were screened towards the EH-315 strain, determining the MIC**_90_** values (see Table 1). Apart from compounds **1** and **17**, all the nitrofuran derivatives show MIC**_90_** ranging from 0.25 to 62.5 µg/mL. In general, compared to the nitrofuran class, the indole one has a lower antifungal activity, with the best compound (**33**) shows a MIC**_90_** value of 1.95 µg/mL. Furthermore, both nitrofuran–indole derivatives **7** and **31** exhibit good antifungal activity, with MIC**_90_** values of 0.125 µg/mL and 3.9 µg/mL, respectively.

A second susceptibility test on *H. capsulatum* (ATCC G217-B strain) was performed using the eighteen compounds that showed MIC**_90_** lower than 15.6 µg/mL in the first screening. As a general trend, the ATCC G217-B strain is less susceptible than EH-315 strain (see Table 2) to the tested molecules. Moreover, AmB and ITZ were tested as reference compounds. Both strains of *H. capsulatum* were drug-sensitive, with results similar to those described in the literature [47,49]. However, it should be considered that the use of these drugs has major adverse effects, such as toxicity and drug interactions [4,14,19]. 

Additionally, for the selected compounds, their MFC values on both the studied strains of *H. capsulatum* were determined. According to the data shown in Table 2, the result on both strains reveals that all the tested compounds have fungicidal activity, with MFCs equal to the MIC**_90_**s, except for **10**, **12**, **14**, and **15**, which require a doubled MIC**_90_** value to kill the fungi. Our findings demonstrate that all tested nitrofuran and indole derivatives had a fungicidal effect against *H. capsulatum* strains, unlike the class of azoles that can have a fungistatic effect against dimorphic fungi [65]. 

On further assays, the cytotoxic effects of the compounds which showed a MIC_90_ equal or lower than 15.6 µg/mL on both strains of *H. capsulatum* were evaluated in the monolayer and the 3D cell cultures for both the A549 and MRC-5 cell lines. Three-dimensional cultures help to recreate an intense physiological environment, such as the in vivo environment [36,38]. The model chosen with agarose gel coating was considered adequate, as it helps to form a semi-solid surface, preventing cells from adhering to the well and, therefore, from being able to interact with each other and form a three-dimensional structure. In addition, the model allows direct contact between cells and the components of the extracellular matrix [36,37,38,71]. 

The developed model presented a regular and compact spheroidal structure, with strong interaction between cells and abundant extracellular matrix. The establishment and characterization of the 3D model is of fundamental importance, as these parameters guarantee the conformity and reproducibility of the technique. Appropriate sizes of spheroids can mimic the in vivo environment. Cultures with diameters smaller than 200 µm will likely not develop all the physiological characteristics, presenting the limited diffusion of molecules, including oxygen, and the accumulation of metabolites. However, spheroids with diameters greater than 600 µm can develop a large area of central necrosis, making it impossible to exchange nutrients and distribute drugs throughout the 3D cell culture [37,72]. The diameters found in this work were around 365.98 ± 66.95 and 574.74 ± 52.03 µm, as established in the literature. In general, strains in 3D culture have an exponential growth phase and, after a certain period, the spheroids reach a linear phase. At this stage, the width of the outer proliferative edge remains constant until the spheroids obtain an adequate size and, from then on, there is a balance between the net cell growth and death rates [72,73,74]. However, it depends on the 3D system used and the cell line [40,75]. In this work, a lower proliferation rate was observed in 3D models than in monolayers, as reported by Gurski et al. [76] and Xu et al. [77], justifying the various stages of development of the spheroid. High cell viability was observed for the 3D models, even after ten days of culture in both methods used (trypan blue and resazurin). Other authors such as Bonnier et al. [78] reported a high cell viability in 3D culture, even having a central region with quiescent cells.

In the study conducted by Friedrich et al. [37], lower cytotoxicity was observed for 3D culture compared to the monolayer treated with the tested compound. This difference between the sensitivity of the monolayer and three-dimensional cultivation may be related to the expression of some genes and proteins [79,80]. Our cytotoxicity results (Table 4 and Table 5) corroborate those reported in the literature; indeed, for most compounds, the CC_50_ values in the three-dimensional models were greater than or equal to those in the monolayer.

To assess the safety of the compounds, the selectivity index (SI) was also calculated. SI indicates the selectivity between the fungal cell and the host [51,81]; in particular, a value of SI higher than 10 indicates a high selectivity for fungal cells [62,82]. Our results reflect low toxicity and high selectivity for most compounds. Though the reference drugs have a low MIC_90_, it is known that there are therapeutic failures and drug interactions in the azole group, in addition to nephrotoxicity [4,14]. On the other hand, polyenes have a narrow therapeutic window, and depending on the patient’s immunological status, the condition can worsen due to their toxic effects [13,83,84]. Therefore, investing in the discovery of new molecules that are effective against histoplasmosis makes the treatment safer and more effective, and expands the range of compounds for treatment.

Although nitrofuran derivatives are known to cause mutagenic and genotoxic effects [16,19], beneficial effects of a drug of this family have already been proven in the treatment of Chagas disease [18]. Additionally, only mild adverse effects were observed in humans [17,18]. Therefore, regulatory agencies such as the Food and Drug Administration (FDA) have approved the commercial use of Nifurtimox in the treatment of Chagas disease [18,85]. Research groups around the world have been carrying out reformulations that make nitrofurans safe for human health and effective against diseases [19,20,25]. 

Compounds **11** (nitrofuran derivative), **32** (indole derivative), and **7** (nitrofuran/indole derivative), i.e., the most active and selective against *H. capsulatum*, were further studied to investigate their mechanism of action. In the literature, there are few reports on the mechanism of action of nitrofuran and indole derivatives on fungal cells. In the study conducted by Kamal et al. [26], nitrofuran derivatives could reduce ergosterol when tested on *Candida parapsilosis*. Furthermore, the same was observed by Gerpe and coworkers [86] against the parasite *Trypanosoma cruzi*. Nitrofurans are believed to accumulate in the cell and interfere with the ergosterol synthesis pathway, inhibiting the squalene epoxidase enzyme [87]. Furthermore, indoles and other nitrogen-based heterocyclic compounds inhibit the conversion of lanosterol into ergosterol [88,89,90]. Our findings corroborate those reported in the literature, as both nitrofuran and indole derivatives can inhibit/reduce fungal membrane ergosterol; however, we cannot predict if they act in the biosynthesis or directly on the membrane, as in the case of AmB, which binds to the membrane ergosterol and can cause a “sponge effect” extracting the ergosterol [91,92,93], and ITZ, which interferes with its synthesis, in both cases with a consequent reduction in the amount of ergosterol in the fungal membrane [51,62]. 

The evaluation of cell wall morphology and damage was performed by calcofluor white, which is a nonspecific fluorophore that binds to chitin in the wall, mainly in the polysaccharides β 1,3 and β 1,4 [94]. Only the nitrofuran derivative **11** caused small flaws in the fungal wall contour. The nitrofuran–indole derivative **7** and the reference drugs (AmB and ITZ) could cause deformations in the fungal cells, increasing their size. This could be related to the osmotic imbalance caused by the rupture of the membrane. The indole derivative **32**, on the other hand, did not differ from the control, similarly to the study conducted by Pooja et al. [89], which reported no disruption or change in cell wall shape when analyzing calcufluor white labeled *C. albicans* cells after indole derivative treatment.

Regarding the ROS induction assay, only the indole derivative **32** and the mixed indole/nitrofuran derivative **7** generated ROS. Hwang et al. [95] showed that indole derivatives could induce ROS production in *C. albicans*. Studies conducted by Singulani et al. [62] and Bila et al. [51] showed that AmB and hydrogen peroxide were ROS inducers. Similar results were found in our work.

There are two pathways of cell death: apoptosis, also called programmed death, which promotes cell stress, activates caspases, and requires energy [96], and necrosis or accidental death, which does not require the use of energy [97,98]. Both nitrofuran and indole derivatives induced deaths mainly using necrosis. This kind of death is possibly related to membrane and cell wall damage, as the disruption of these structures irreversibly compromises cell homeostasis.

## 5. Conclusions

Our study identified nitrofuran and indole derivatives as anti-*Histoplasma capsulatum* agents, with fungicidal activity, reaching an MFC value of 0.122 µg/mL for the most active compound with the nitrofuran and the indole moieties (**7**). The evaluation of their toxicity shows that the 3D model, capable of simulating the physiology of an in vivo environment, was less susceptible to the compounds than cell monolayers of lung cells. The analysis of the mechanisms of action of the most active and selective compounds (**7**, **11**, and **32**) showed production of ROS, a reduction in the amount of ergosterol in the fungal membrane, disruption of the cell wall, and induction of death using necrosis. These results show that indole and nitrofuran derivatives are promising molecular scaffolds for the development of original anti-histoplasmosis drug candidates.

## Figures and Tables

**Figure 1 pharmaceutics-14-01043-f001:**
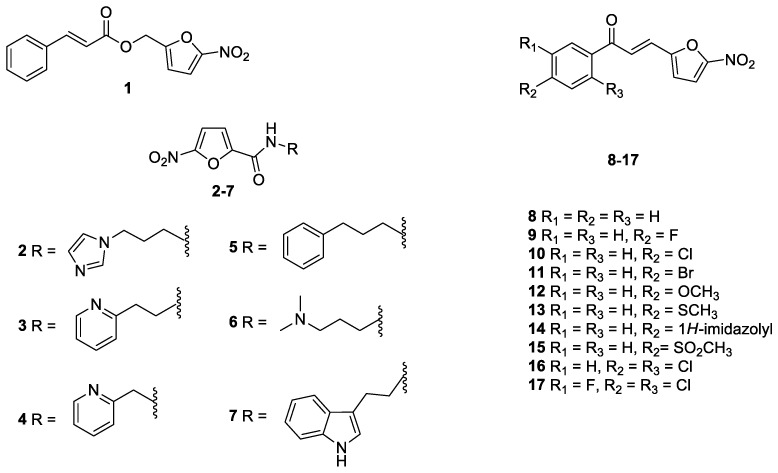
Nitrofuran derivatives **1**–**17** studied in this work.

**Figure 2 pharmaceutics-14-01043-f002:**
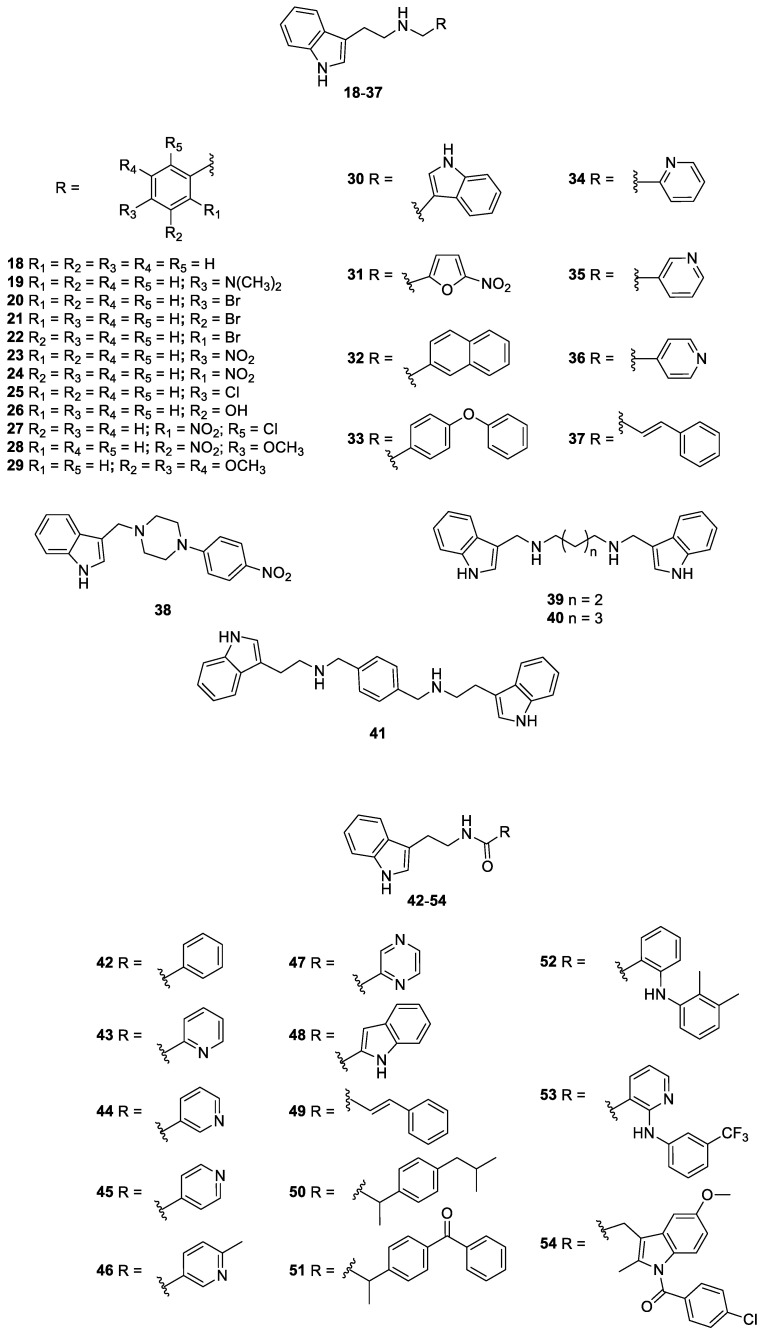
Indole derivatives **18**–**54** studied in this work.

**Figure 3 pharmaceutics-14-01043-f003:**
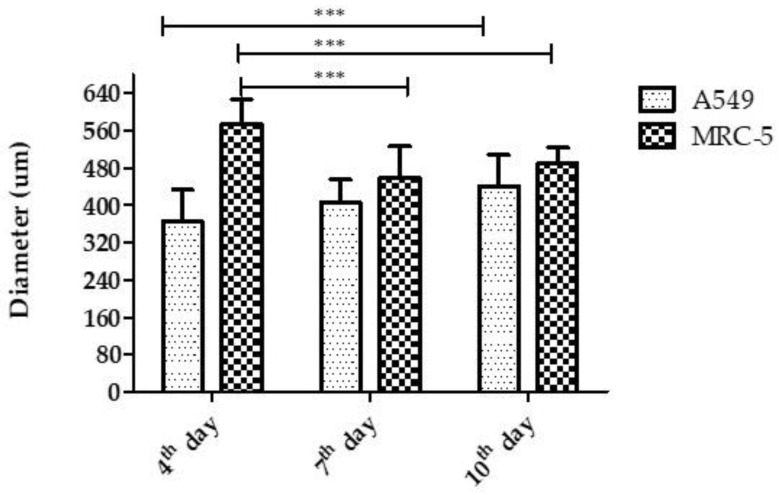
Diameter of the spheroids of the A549 and MRC-5 cell lines on the fourth, seventh, and tenth day of formation. Values expressed as mean and standard deviation. Statistical differences between the A549 cell line from the fourth day in relation to the tenth day and for the MRC-5 cell line from the fourth day in relation to the seventh and tenth day. *** *p* < 0.001.

**Figure 4 pharmaceutics-14-01043-f004:**
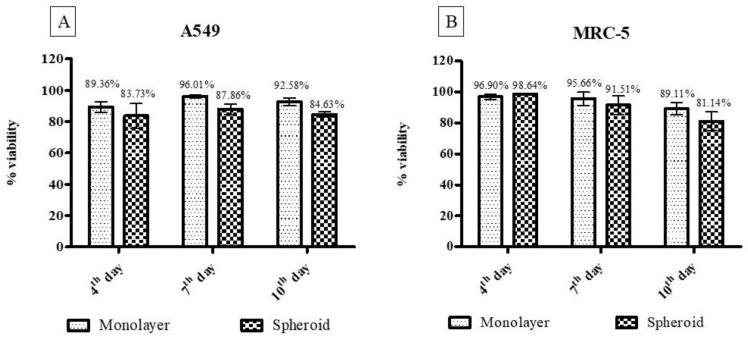
Cell viability results from the trypan blue method. In Figure (**A**), the comparative values between the monolayer and three-dimensional model for the A549 cell line are shown, and in (**B**), the same comparative results for the MRC-5 cell line are shown. There was no statistically significant difference.

**Figure 5 pharmaceutics-14-01043-f005:**
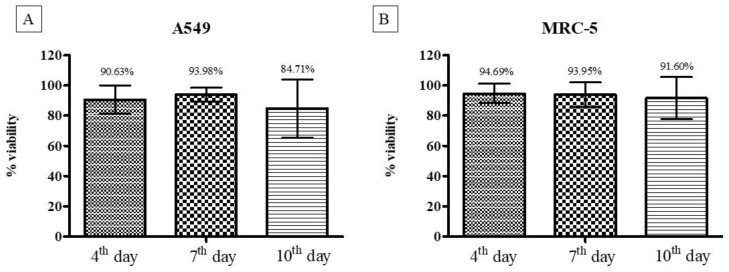
Cell viability using the resazurin method for the three-dimensional model of cell lines A549 (**A**) and MRC-5 (**B**). In both cell lines, the viability of the three-dimensional culture remained stable after ten days of culture. There was no statistically significant difference.

**Figure 6 pharmaceutics-14-01043-f006:**
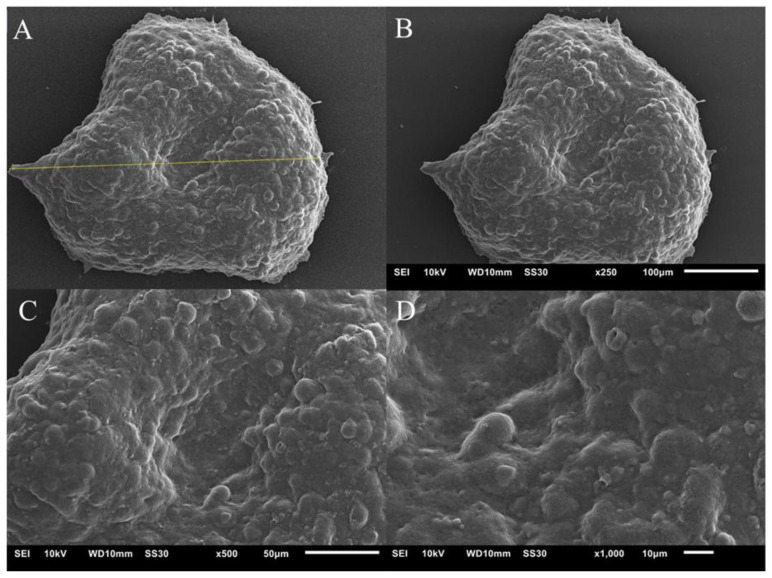
Image of the spheroid of the A549 cell line. (**A**)—the yellow line indicates the diameter of the spheroid, measuring 389.25 µm; (**B**)—spheroid on ×250 objective; (**C**)—image on ×500 objective and (**D**)—×1000 objective.

**Figure 7 pharmaceutics-14-01043-f007:**
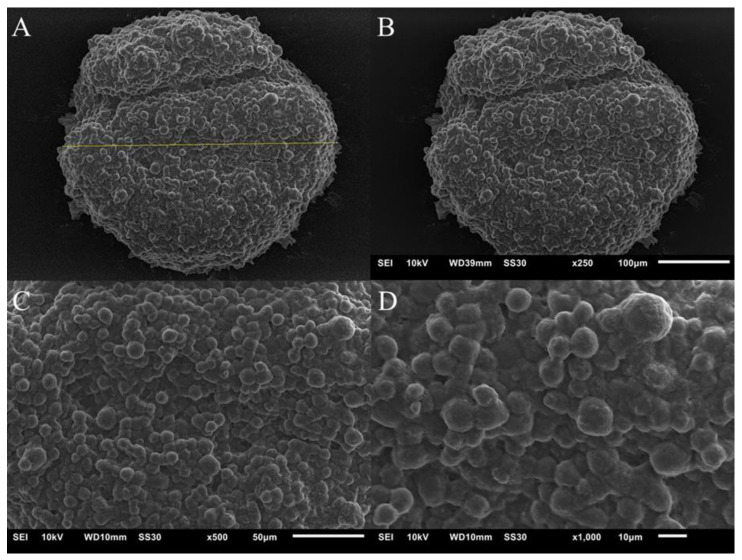
Image of the spheroid of the MRC-5 cell line. (**A**)—the yellow line indicates the diameter of the spheroid, measuring 389.33 µm; (**B**)—spheroid on ×250 objective; (**C**)—image on ×500 objective and (**D**)—×1000 objective.

**Figure 8 pharmaceutics-14-01043-f008:**
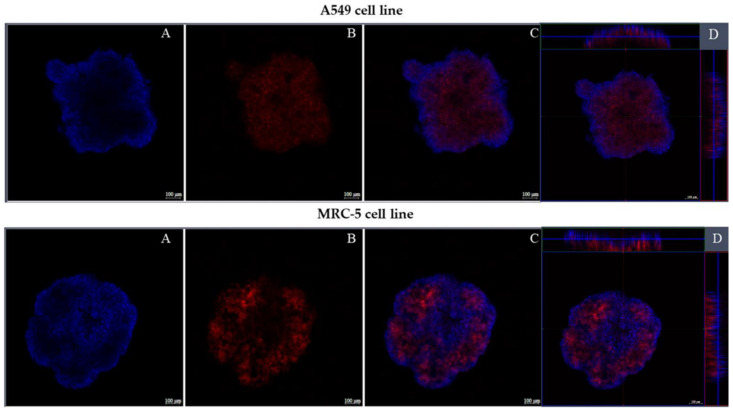
Fluorescence of three-dimensional cultures of lines A549 and MRC-5. (**A**)—the blue marking shows the nucleus of the cells; (**B**)—in red the cytoplasm of the cells; (**C**)—merge of the red and blue markings and (**D**)—shows the depth of the three-dimensional crops.

**Figure 9 pharmaceutics-14-01043-f009:**
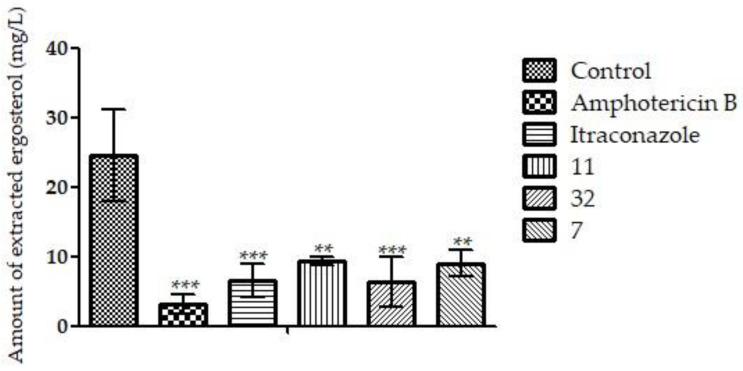
The graphic shows the amount of extracted ergosterol of fungal H. capsulatum EH-315 treated with **11**, **32**, **7**, ITZ, and AmB. In the treatment with compounds **11**, **32**, and **7**, the amounts of sterols extracted were reduced, indicating that these compounds may act on the ergosterol synthesis or chain. AmB and ITZ also reduced the amount of steroids due to their mechanisms of action in the membrane and on the synthesis chain, respectively. ** *p* < 0.01; *** *p* < 0.001.

**Figure 10 pharmaceutics-14-01043-f010:**
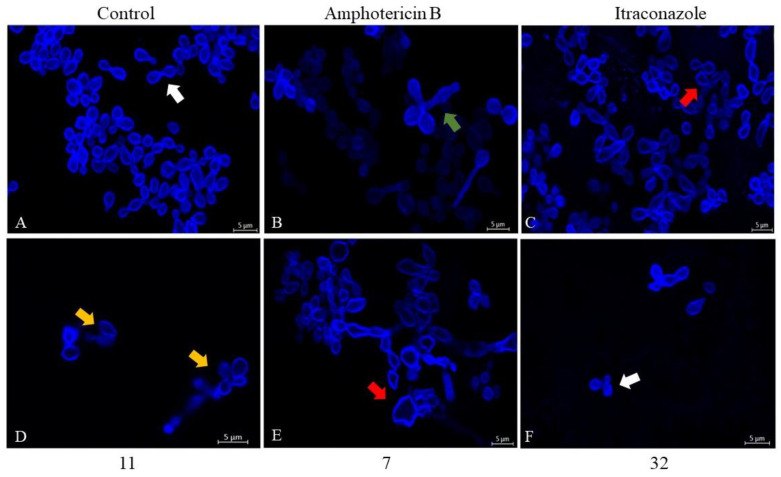
Confocal laser scanning microscopy images (CLSM) of fungal H. capsulatum EH-315 treated with **11**, **7**, **32**, ITZ, and AmB. In CLSM images, the cell wall was stained with calcofluor white. Cells treated with compound **11** (**D**) showed small cell wall faults (orange arrow). However, cells treated with AmB (**B**) showed prolongations (green arrow). ITZ (**C**) and **7** (**E**) caused irregularities in cell shape (red arrows). Both the cells without treatment (**A**) and those treated with the compound **32** (**F**) had a continuous and regular wall (white arrow).

**Figure 11 pharmaceutics-14-01043-f011:**
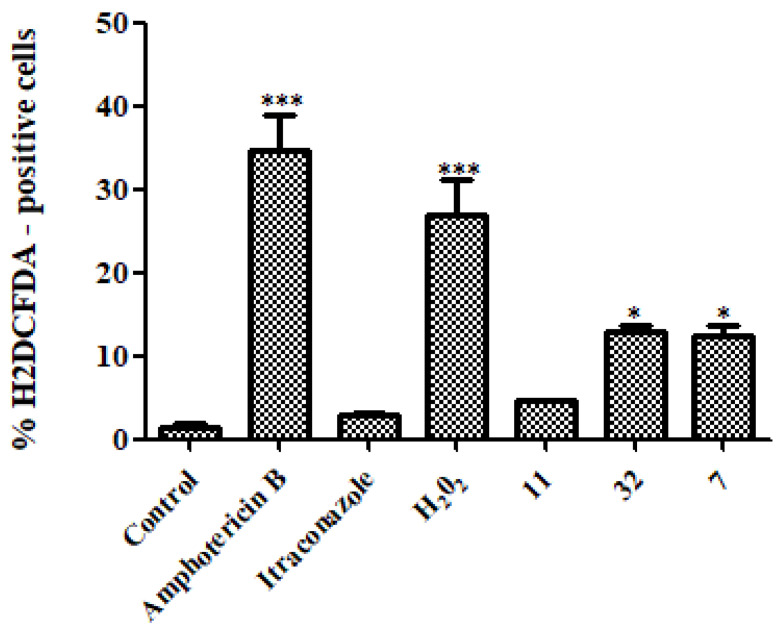
Measurement of ROS production after treatment of H. capsulatum EH-315 with **11**, **32**, **7**, AmB, ITZ, and hydrogen peroxide. The drug AmB, hydrogen peroxide, and compounds **32** and **7** induced ROS formation when compared to the control without treatment. Treatment with ITZ did not induce ROS formation when compared to the control. * *p* < 0.05; *** *p* < 0.001.

**Figure 12 pharmaceutics-14-01043-f012:**
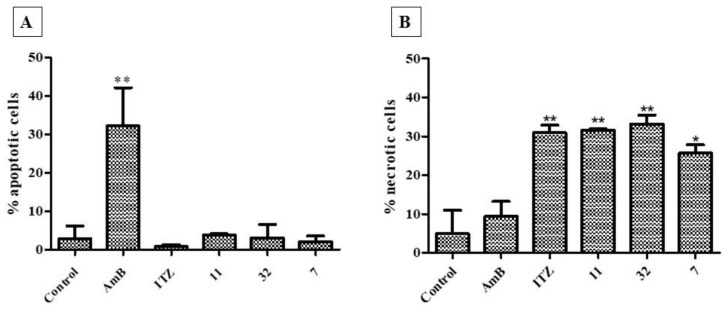
Mechanism of death due to apoptosis (**A**) and necrosis (**B**) induced after treatment of H. capsulatum EH-315 with **7**, **11**, and **32**, in addition to the drug controls AmB and ITZ, compared with the untreated control. The AmB induced cell death by apoptosis, whereas ITZ, **7**, **11**, and **32** induced death by necrosis. * *p* < 0.05; ** *p* < 0.01.

**Table 1 pharmaceutics-14-01043-t001:** Antifungal activity (MIC_90_) of fifty-four nitrofuran and indole derivatives against the *H. capsulatum* EH-315 strain.

Compound	Concentration (µg/mL)	Compound	Concentration (µg/mL)	Compound	Concentration (µg/mL)
**1**	≥125	**20**	7.8	**39**	7.8
**2**	15.6	**21**	7.8	**40**	62.5
**3**	1.95	**22**	31.25	**41**	7.8
**4**	0.98	**23**	31.25	**42**	31.25
**5**	0.25	**24**	62.5	**43**	62.5
**6**	62.5	**25**	≥125	**44**	125
**7**	0.122	**26**	≥125	**45**	62.5
**8**	31.25	**27**	≥125	**46**	31.25
**9**	0.98	**28**	31.25	**47**	125
**10**	7.8	**29**	31.25	**48**	≥125
**11**	0.25	**30**	15.6	**49**	125
**12**	7.8	**31**	3.9	**50**	3.9
**13**	62.5	**32**	3.9	**51**	31.25
**14**	3.9	**33**	1.95	**52**	≥125
**15**	7.8	**34**	≥125	**53**	≥125
**16**	31.25	**35**	125	**54**	≥125
**17**	≥125	**36**	≥125		
**18**	31.25	**37**	15.6		
**19**	15.6	**38**	≥125		

**Table 2 pharmaceutics-14-01043-t002:** MIC_90_ and MFC values for *H. capsulatum* EH-315 and ATCC G217-B strains, for the eighteen most potent compounds in the initial screening.

		EH-315 (µg/mL)		ATCC G217-B (µg/mL)	
Compounds	Derivate	MIC_90_	MFC	MIC_90_	MFC
**3**	Nitrofuran	1.95	1.95	3.9	3.9
**4**	Nitrofuran	0.98	0.98	3.9	3.9
**5**	Nitrofuran	0.24	0.24	1.95	1.95
**7**	Nitrofuran/Indole	0.122	0.122	0.98	0.98
**9**	Nitrofuran	0.98	0.98	3.9	3.9
**10**	Nitrofuran	7.81	7.81	7.81	15.62
**11**	Nitrofuran	0.24	0.24	0.48	0.48
**12**	Nitrofuran	7.81	7.81	7.81	15.62
**14**	Nitrofuran	3.90	7.81	7.81	7.81
**15**	Nitrofuran	7.81	15.62	>250	>250
**20**	Indole	7.81	7.81	7.81	7.81
**21**	Indole	7.81	7.81	7.81	7.81
**31**	Nitrofuran/indole	3.90	3.90	31.25	31.25
**32**	Indole	3.90	3.90	3.90	3.90
**33**	Indole	1.95	1.95	3.90	3.90
**39**	Indole	7.81	7.81	15.62	15.62
**41**	Indole	7.81	7.81	61.25	61.25
**50**	Indole	3.90	3.90	31.25	31.25
**AmB**	Polyene	0.06	-	0.03	-
**ITZ**	Azole	0.125	-	0.007	-

**Table 3 pharmaceutics-14-01043-t003:** Cell quantification results for both the monolayer and three-dimensional models using the trypan blue method.

Cell Line	Model	Mean—Fourth Day	Mean—Seventh Day	Mean—Tenth Day
A549	Monolayer	5.23 *×* 10^4^ ± 7.56 *×* 10^3^	8.50 *×* 10^4^ ± 2.73 *×* 10^4^	1.03 *×* 10^5^ ± 2.08 *×* 10^3^
	Spheroid	8.33 *×* 10^3^ ± 2.36 *×* 10^3^	1.23 *×* 10^4^ ± 1.53 *×* 10^3^	1.25 *×* 10^4^ ± 2.08 *×* 10^3^
MRC-5	Monolayer	1.91 *×* 10^5^ ± 1.53 *×* 10^4^	3.56 *×* 10^5^ ± 3.96 *×* 10^4^ (*)	3.06 *×* 10^5^ ± 3.58 *×* 10^4^
	Spheroid	2.69 *×* 10^3^ ± 6.67 *×* 10^2^	4.25 *×* 10^4^ ± 3.75 *×* 10^3^ (***)	3.13 *×* 10^4^ ± 7.60 *×* 10^3^ (**)

(*) *p* < 0.05; (**) *p* < 0.01; (***) *p* < 0.001. Values are expressed as the mean and standard deviation.

**Table 4 pharmaceutics-14-01043-t004:** Values of CC_50_ (µg/mL) and SI for the A549 cell line in the monolayer and the 3D model treated with nitrofuran and indole derivatives.

	A549					
	Monolayer			Spheroid		
		EH-315	ATCC G217-B		EH-315	ATCC G217-B
Compounds	CC_50_ (µg/mL)	SI	SI	CC_50_ (µg/mL)	SI	SI
**3**	231.2	118.56	59.28	>250	>128.20	>64.10
**4**	29.50	30.10	7.56	62.96	64.24	16.14
**5**	30.11	125.45	15.44	43.01	179.20	22.05
**7**	8.58	70.32	8.75	>250	>2049.18	>255.1
**9**	23.41	23.88	6.00	17.25	17.60	4.42
**10**	38.67	4.95	4.95	32.97	4.22	4.22
**11**	12.24	50.16	25.50	56.06	229.75	116.79
**12**	>250	>32.01	>32.01	104.8	13.41	13.41
**14**	33.44	8.83	4.4	27.61	7.07	3.53
**20**	81.41	10.42	10.42	81.41	10.42	10.42
**21**	40.00	5.12	5.12	40.00	5.12	5.12
**32**	81.41	20.87	20.87	81.41	20.76	20.76
**33**	12.26	6.28	3.14	13.22	6.77	1.69
**39**	42.79	5.47	2.73	60.5	7.74	3.87

CC_50_—concentration that inhibits cell proliferation by 50%; SI—selectivity index (CC_50_/MIC_90_).

**Table 5 pharmaceutics-14-01043-t005:** Values of CC_50_ (µg/mL) and SI for the MRC-5 cell line in the monolayer and the 3D model treated with nitrofuran and indole derivatives.

	MRC-5					
	Monolayer			Spheroid		
		EH-315	ATCC G217-B		EH-315	ATCC G217-B
Compounds	CC_50_ (µg/mL)	SI	SI	CC_50_ (µg/mL)	SI	SI
**3**	>250	>128.20	>64.10	>250	>128.20	>64.10
**64**	135.2	137.95	34.66	>250	> 255.10	>64.10
**5**	39.76	165.66	20.38	81.41	339.20	41.74
**7**	71.24	583.93	72.69	73.50	602.45	75.00
**9**	47.23	48.19	12.11	169.6	173.06	43.48
**10**	215.4	27.58	27.58	57.92	7.41	7.41
**11**	62.56	256.39	130.33	67.79	277.82	141.22
**12**	228.0	29.19	29.19	76.12	9.74	9.74
**14**	26.8	6.87	3.43	7.68	1.96	0.98
**20**	20.39	2.61	2.61	25.35	3.57	3.57
**21**	33.77	4.32	4.32	52.34	6.70	6.70
**32**	24.7	6.33	6.33	26.14	6.70	6.70
**33**	18.74	9.56	4.78	14.03	7.19	3.59
**39**	40.00	5.12	2.56	40.00	5.12	2.56

CC_50_—concentration that inhibits cell proliferation by 50%; SI—selectivity index (CC_50_/MIC_90_).

## Data Availability

Not applicable.

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
