# Peer review of "Evaluation of the Anti-Histoplasma capsulatum Activity of Indole and Nitrofuran Derivatives and Their Pharmacological Safety in Three-Dimensional Cell Cultures"

_pharmaceutics, 2022, doi:10.3390/pharmaceutics14051043_

Round 1
Reviewer 1 Report
From fifty-four nitrofurans and indoles, authors screened out three of them with higher antifungal activity against Histoplasma capsulatum and lower toxicity against a lung epithelial cell line and a pulmonary fibroblast cell line. Effects of these three compounds on ergosterol content, cell morphology, ROS production, and cell dearth were tested. Overall, the manuscript provides some useful information on biological activity of tested nitrofurans and indoles. However, it still needs improvement.
- Results about Table3, Figure 3, 4 5and 6 are the base to start the main work in this manuscript but are not very closely related the main topic of the manuscript. The presentation of detailed comparison results between the lung epithelial cell line and the pulmonary fibroblast cell line go too far away from the key objectives.
- For determining ergosterol: “The analyzes were carried out in a spectrophotometer at a wavelength of 281 nm with an ultraviolet (UV) spectrum. Standard curves were prepared with 95% purity ergosterol at concentrations ranging from 75 to 10 µg/mL”
Comments: Usually, fungal cells have several different sterols including ergosterol. To accurately quantify each of them, Liquid chromatography separation followed with detection with Mass spectra is the most frequently used method. It is not sure how reliable the method used in the manuscript is. Please add a reference(s). Please also add the information on where the ergosterol standard is from.
- In 464, “The drugs AmB and ITZ also reduced the amount of ergosterol extracted, as expected ”.
Comments: AmB binds to ergosterol but should not inhibit ergosterol synthesis based on its mode of action. Reduction of ergosterol upon AmB treatment is not expected. Have the similar results been previously reported? It could be caused by the method of ergosterol analysis? Please discuss the reason.
- In abstract, from line 35 to 38, “Furthemore, 7 and 32 induce reactive oxygen species (ROS) formation higher than 11 and the control. Finally, the cytotoxicity was performed on cell monolayers and three-dimensional (3D) cultures forms. Thecytotoxicity assays in the 3D model showed a lower toxicity of the compounds than those performed on cell monolayers”.
Comments: Based on these words, readers could not figure out what cell monolayers and three-dimensional (3D) cultures are.
- Line 477, "To check for possible wall damage, the fluorophore Calcofluor White was used which stains chitin and cell wall cellulose".
Comments: Fungal cell wall is mainly composed of glucan, chitin and mannoprotein. Cellulose in fungal cell wall is not common. In addition, Calcofluor stainig are generally used to observe the morphology of fungal cell. Damage in cell wall is not easily visualized by this staining. Thus, figure 10 can only shows the effects of tested compounds on cell morphology rather than cell wall.
- The underlined words in the following sentences need to check to grammar:
1) Line 112-113, strains were subsequently subculture in the Ham’s F-12 Nutrient Mixture medium
2) Line 127, Antifungal drugs stock 127
3) 3.3.2. Verification wall damage
Author Response
From fifty-four nitrofurans and indoles, authors screened out three of them with higher antifungal activity against Histoplasma capsulatum and lower toxicity against a lung epithelial cell line and a pulmonary fibroblast cell line. Effects of these three compounds on ergosterol content, cell morphology, ROS production, and cell dearth were tested. Overall, the manuscript provides some useful information on biological activity of tested nitrofurans and indoles. However, it still needs improvement.
- Results about Table3, Figure 3, 4 5and 6 are the base to start the main work in this manuscript but are not very closely related the main topic of the manuscript. The presentation of detailed comparison results between the lung epithelial cell line and the pulmonary fibroblast cell line go too far away from the key objectives.
The authors thank the reviewer for all suggestions. The objective of demonstrating the characterization of three-dimensional cultures was a way of simplifying how we performed the cytotoxicity experiments and clarifying the methodology. In addition, the intention was to demonstrate that the results found are reliable and perishable for reproduction, including for testing on antifungals, since the 3D model was not explored to assess the toxicity of new antifungal molecules, only on antitumor molecules and antibiotics. In addition, the 3D model features tissue-like cell-cell and cell-matrix interactions, which makes this model more effective for testing new compounds (Belfiore, et al., 2021, https://doi.org/10.1016/j.ejps.2021.105876; Habanjar, et al., 2021, https://doi.org/10.3390/ijms222212200; Edmondson, 2014, https://doi.org/10.1089/adt.2014.573; Friedrich, et al., 2009, https://doi.org/10.1038/nprot.2008.226).
- For determining ergosterol: “The analyzes were carried out in a spectrophotometer at a wavelength of 281 nm with an ultraviolet (UV) spectrum. Standard curves were prepared with 95% purity ergosterol at concentrations ranging from 75 to 10 µg/mL”
Comments: Usually, fungal cells have several different sterols including ergosterol. To accurately quantify each of them, Liquid chromatography separation followed with detection with Mass spectra is the most frequently used method. It is not sure how reliable the method used in the manuscript is. Please add a reference(s). Please also add the information on where the ergosterol standard is from.
We appreciate the observation raised. All changes are highlighted in green. In fact, fungal cells have several sterols in their entirety, however, the most abundant is ergosterol. Therefore, it makes it an important target of antifungal action. We know that by liquid chromatography coupled to the mass spectrum the results would be more accurate, however, several authors use this simpler technique with spectrophotometer reading obtaining interesting results (Arthington-Skaggs, B. A., et al. 1999, https://doi.org/10.1128/JCM.37.10.3332-3337.1999; Jothi, R., et al., 2021, https://doi.org/10.1038/s41598-021-00485-2; Arthington-Skaggs, B. A., et al., 2000, https://doi.org/10.1128/AAC.44.8.2081-2085.2000, Bila, N. M., et al., 2021, https://doi.org/10.3389/fcimb.2021.679470; Bvumbi, C., et al., (2021), https://doi.org/10.1155/2021/8856147). We added some more references that use the same technique as us, for the quantification of ergosterol and more information about standard ergosterol “Pag 9, lines 272 and 273”.
In 464, “The drugs AmB and ITZ also reduced the amount of ergosterol extracted, as expected”.
Comments: AmB binds to ergosterol but should not inhibit ergosterol synthesis based on its mode of action. Reduction of ergosterol upon AmB treatment is not expected. Have the similar results been previously reported? It could be caused by the method of ergosterol analysis? Please discuss the reason.
We thank the reviewer and agree with the statement, amphotericin B binds to ergosterol causing pores and thus causes ion extravasation, it really does not act on biosynthesis (Pag. 2, lines 57 and 58; Pag. 20, lines 446 and 447; Pag. 25, lines 576 and 578). Several authors have demonstrated the sponge effect of AmB, where it binds to ergosterol in the membrane and extracts it (Anderson T. M., et al., 2014, https://doi.org/10.1038/nchembio.1496; Guo, X., et al., 2021, https://doi.org/10.1021/acscentsci.1c00148; Lewandowska, A., et al., 2021, https://doi.org/10.1038/s41594-021-00685-4). The same authors demonstrate the reduction of ergosterol extracted in yeasts treated with this drug. In the manuscript we added a statement about this “Pag. 25, lines 576 to 578”.
In abstract, from line 35 to 38, “Furthemore, 7 and 32 induce reactive oxygen species (ROS) formation higher than 11 and the control. Finally, the cytotoxicity was performed on cell monolayers and three-dimensional (3D) cultures forms. The cytotoxicity assays in the 3D model showed a lower toxicity of the compounds than those performed on cell monolayers”.
Comments: Based on these words, readers could not figure out what cell monolayers and three-dimensional (3D) cultures are.
The sentence was modified for better comprehension “Pag. 1, lines 36 to 38”.
Line 477, "To check for possible wall damage, the fluorophore Calcofluor White was used which stains chitin and cell wall cellulose".
Comments: Fungal cell wall is mainly composed of glucan, chitin and mannoprotein. Cellulose in fungal cell wall is not common. In addition, Calcofluor stainig are generally used to observe the morphology of fungal cell. Damage in cell wall is not easily visualized by this staining. Thus, figure 10 can only shows the effects of tested compounds on cell morphology rather than cell wall.
We thank the reviewer and agree with the statement, we think it is better to remove "cellulose" from the text to facilitate understanding “Pag. 20, lines 450 and 451”. Regarding the Calcofluor staining, it is used to observe the morphology of the cells, we also carried out this evaluation in the manuscript, however, in the literature there are already works that use this methodology to evaluate cell wall damage, even though it is not easy to visualize (Victoria S. G., Pravin K., Komath S.S., 2010, https://doi.org/10.1099/mic.0.039628-0; Pooja P. P., 2014, https://doi.org/10.1016/j.ejmech.2014.04.063; Bila N.M., et al., 2021, https://doi.org/10.3389/fcimb.2021.679470). Our compounds showed more conformational wall deformities, except for compound 11 which apparently caused flaws in the cell wall contour. We made small changes to the manuscript to better suit it “Pag. 20 lines 449 to 451; Pag. 25, lines 580 to 582”.
The underlined words in the following sentences need to check to grammar:
1) Line 112-113, strains were subsequently subculture in the Ham’s F-12 Nutrient Mixture medium
2) Line 127, Antifungal drugs stock 127
3) 3.3.2. Verification wall damage
All sentences have been modified “Pag. 5, line 123; Pag. 6, lines 138 and 139; Pag. 9, line 274 and 275”
Reviewer 2 Report
In the present study, antifungal activity of fifty-four nitrofurans and indoles were tested against H. capsulatum, and 7, 11 and 32 were screened form them. The mechanism of action of them was investigated. This work will fetch interest among researchers in relevant field.
Minor comments to the authors are as follows:
In “Determination of minimum fungicide concentration (MFC)”, 100 µL aliquots of the content of the wells were removed and transferred to agar plates, are these 100 µL aliquots mixture of compound and fungal cells in MIC assay?
Please unify the description of “4th, 7th and 10th day”, “the fourth, seventh, and tenth days” .
In Ergosterol dosage assay, The compounds 11 (0.122 µg/mL), 32 (1.95 µg/mL) and 7 (0.06 µg/mL), and control AmB (0.03 µg/mL) and ITZ (0.06 µg/mL) were used. Why used these concentrations?
What is the incubation time in ROS assay?
Data in Table 1 is MIC90? Title of Table 1 sholud be “Antifungal activity (MIC90) of fifty-four nitrofuran and indole derivatives against the H. capsulatum EH-315 strain”.
Please move “Values expressed as mean and deviation.” in Line 375 to 2.6. Statistical analysis section.
Please add “3.2.3” before “viability by resazurin colorimetric method” in Line 395.
Please Statistically analyzes data in Fig 4 and 5.
Please add “significantly” before “reduce the extracted ergosterol” in Line 463; also in Line 464.
In Verification wall damage assay, except wall damage, Is cells amount difference among treatments?
Please delete “AmB – Amphotericin B; ITZ – itraconazole” in line 503.
Please indicate “A” for apoptosis fig and “B” for necrosis fig in Fig 12.
Author Response
In “Determination of minimum fungicide concentration (MFC)”, 100 µL aliquots of the content of the wells were removed and transferred to agar plates, are these 100 µL aliquots mixture of compound and fungal cells in MIC assay?
The authors thank the reviewer for all suggestions. All changes made to the manuscript are highlighted in pink. Aliquots were taken from the wells of microdilution plates that contained different concentrations of compounds treating the same concentration of fungal inoculum. As it is a methodology widely described by several authors Costa-Orlandi and collaborators, 2020 (https://doi.org/10.3389/fmicb.2020.01154); Soares and collaborators, 2014 (//doi.org/10.1155/2014/957860); Ghannoum, M., Isham, N., & Long, L., 2015 (https://doi.org/10.1128/AAC.00992-15); Brilhante and collaborators, 2014 (https://doi.org/10.1093/mmy/myt027).
Please unify the description of “4th, 7th, and 10th day”, “the fourth, seventh, and tenth days”.
The writing was unified “Pag. 7, Line 190; Pag. 13, Lines 354 and 355; Pag. 13, Table 3”.
In Ergosterol dosage assay, The compounds 11 (0.122 µg/mL), 32 (1.95 µg/mL) and 7 (0.06 µg/mL), and control AmB (0.03 µg/mL) and ITZ (0.06 µg/mL) were used. Why used these concentrations?
Sub-inhibitory concentrations of compounds and drugs were used. Since concentrations equal to or greater than the MIC would not allow the fungus to grow, it would not be possible to quantify membrane sterols. The sub-inhibitory concentration allows the observation of some functional effects of the fungus without inhibiting its growth.
What is the incubation time in ROS assay?
The incubation time was added in the text. Please see Pag.10, line 296. As recommended by the reagent manufacturer (H2DCFDA), it was incubated for 4h.
Data in Table 1 is MIC90? Title of Table 1 should be “Antifungal activity (MIC90) of fifty-four nitrofuran and indole derivatives against the H. capsulatum EH-315 strain”.
Yes, data in Table 1 correspond to MIC90. The table title has been modified “Pag. 11, Line 327”.
Please move “Values expressed as mean and deviation.” in Line 375 to 2.6. Statistical analysis section.
The phrase was taken from line 375 and added to topic 2.6. “Pag. 10, Lines 318 and 319”.
Please add “3.2.3” before “viability by resazurin colorimetric method” in Line 395.
The modification was conducted. Please see “Pag. 14, line 376”.
Please Statistically analyzes data in Fig 4 and 5.
Both figures 4 and 5 were subjected to statistical analysis, however, there was no significant difference. These statements were included in the legend of the figures.
Please add “significantly” before “reduce the extracted ergosterol” in Line 463; also in Line 464.
The word has been added “Pag. 19, lines 437 and 438”.
In Verification wall damage assay, except wall damage, Is cells amount difference among treatments?
The experiments were performed with the same cell concentration (1×106 cells/mL) and sub-inhibitory concentrations of compounds and drugs. However, we did not evaluate this variable, although compounds 7 and 32 apparently caused a reduction in the number of cells.
Please delete “AmB – Amphotericin B; ITZ – itraconazole” in line 503.
Abbreviations and names have been deleted from line 503.
Please indicate “A” for apoptosis fig and “B” for necrosis fig in Fig 12.
The letters A and B were indicated in the figure and legend “Pag 22 – Figure 12”.
Reviewer 3 Report
The authors describe in this manuscript the in vitro evaluation of almost 60 derivatives, already described by their research team, against Histoplasma capsulatum. One of the strong points of the manuscript is the conduction of in vitro cytotoxicity assays in both 2D and 3D culture forms. This work identified 3 hit molecules in nitrofuran, indole and nitrofuran/indole series, respectively.
I have some remarks and suggestions, mainly typos and rewording suggestions, to bring to the attention of the authors:
- Line 28: please remove “The” before “Compounds with…”
- Line 34: to remain consistent with the rest of the abstract written in the past tense, please write “damaged the cell wall”
- Line 36: same remark, please write “induced”
- Line 36: please remove “the” before “control”
- Line 37, rewording suggestion: “cytotoxicity was measures”, and not performed
- Line 37: please remove “The” before “cytotoxicity assays”
- Line 48: a bibliographic reference at the end of the sentence about the annual incidence of histoplasmosis is expected
- Line 48: In this introductory paragraph, a few words recalling the pathophysiology of the disease would be welcome.
- Line 54, rewording suggestion: “The main drawbacks related to AmB are…”
- Line 55: more details on the nature of amphotericin B toxicities would be welcome
- Line 56: consider rewording this last sentence, as “problems” is a rather vague term
- Line 58: the term "derivatives" may be preferable to "derivates"; at the discretion of the authors
- Line 59: nitrofuran derivatives such as nifurtimox are also used in human medicine; this should be mentioned more evidently
- Line 60: same remark concerning the “toxicity-related” issues of nitrofurans that should be more clearly stated
- Line 67: please remove the coma before “such as cancer”
- Line 70: please remove “models”
- Line 73: the bibliographic reference corresponding to the articles describing the tested compounds should be mentioned here, in addition to being included in the experimental part
- Line 92: please write “the in vivo environment”
- Line 113: please write “subcultured”
- Lines 111, 115 and 116: please replace the coma by a dot in “0,1%”, 0,1%” and “0,6%”, respectively
- Line 116: please write “0.008% L-cysteine” without a space between the number and the percent symbol, and without a capital letter at cysteine.
- Line 125: the term "derivatives" may be preferable to "derivates"
- Line 137: please remove “the” before “Trypan blue”
- Line 138: please replace the “x” in “1 x 106” by a proper multiplication symbol, as it is the case right after
- Line 171: please reconsider rewording this sentence as it is not grammatically correct
- Lines 257 and 260: please consider rephrasing these sentences so that they do not begin with a number, or consider writing these numbers in words.
- Line 300: please write “2 h” and not “2 hours”
- Line 301: please check the possible extra space before “1x PBS”
- Line 314: for all the results given in µg/mL, wouldn’t it be more relevant to express them in molar concentration to overcome the variation in molecular weight between each molecule? This would sound more rigorous
- Line 343-343: please add space before and after the “<” symbol, as it is done afterwards
- Line 372-373: same remark, please add space before and after the “<” symbol
- Line 376: same remark, please add space before and after the “<” symbol
- Table 3: please replace all the “x” by a proper multiplication symbol
- Line 395: please add a capital letter at “Viability”
- Line 442: why the cytotoxicity results are given in inhibitory concentrations (IC50) and not cytotoxic concentrations (CC50) if it is the cytotoxicity of these compounds that is measured? This point needs to be clarified.
- Tables 4 and 5: as several cytotoxicity values are “greater than” a given value, the resulting selectivity indices are also “greater than” the ratio calculated. Please add the “>” symbol before the SI values concerned (6 occurrences in Table 4; 8 occurrences in Table 5)
- Line 480: please remove “The” before “Compound 11”
- Line 481: please remove “the” before “compound 7”
- Line 483: please remove “the” before “compound 32”
- Line 506, rewording suggestion: “Although effective, therapeutic options for treating histoplasmosis are limited and with high risks of toxicity for the patient. Therefore, …”
- Line 516: please remove “the” before “compounds 1 and 17”
- Line 526-527: please reconsider rewording as this sentence, although understandable, is quite unclear
- Line 555: please write “the 3D cell culture”
- Line 572: please consider replacing “IC50” by “CC50”
- Line 577: a comparison to reference compounds would be expected in the discussion
- Discussion part: one of the well-known toxicities of nitro-containing molecules such as nifurtimox is their mutagenicity and genotoxicity. The corresponding in vitro evaluations (Ames test and comet assay) on compounds 7 and 11 would be of great value in this study. If these tests can’t be undertaken as a part of this work, this topic should still be discussed in depth.
- Line 625-626: this statement is very premature (as no extensive toxicity, physicochemical, stability and especially in vivo studies were undertaken on these molecules) and should be reworded. Suggestion: … derivatives are promising molecular scaffolds for the development of original anti-histoplasmosis drug candidates”
- Line 652: bibliographical references should be reworked in depth to be all formulated in a homogeneous way in accordance with the citation style of the Multidisciplinary Digital Publishing Institute. See the publisher's website.
Author Response
I have some remarks and suggestions, mainly typos and rewording suggestions, to bring to the attention of the authors:
- Line 28: please remove “The” before “Compounds with…”
The authors thank the reviewer for all suggestions. All changes made to the manuscript are highlighted in blue. The “the” has been removed “Pag. 1, line 28”.
- Line 34: to remain consistent with the rest of the abstract written in the past tense, please write “damaged the cell wall”
The change was made “Pag 1, line 34”.
- Line 36: same remark, please write “induced”
The change was made “Pag. 1, line 35”.
- Line 36: please remove “the” before “control”
The word has been removed “Pag. 1, line 36”.
- Line 37, rewording suggestion: “cytotoxicity was measures”, and not performed
The suggestion was accepted “Pag. 1, line 36”.
- Line 37: please remove “The” before “cytotoxicity assays”
The word has been removed “Pag. 1, line 38”.
- Line 48: a bibliographic reference at the end of the sentence about the annual incidence of histoplasmosis is expected
References have been added to the end of the sentence “Pag. 2, line 48”.
- Line 48: In this introductory paragraph, a few words recalling the pathophysiology of the disease would be welcome.
Added more information about the pathology of the disease “Pag. 2, lines 50,51 and 52”.
- Line 54, rewording suggestion: “The main drawbacks related to AmB are…”
The suggestion was accepted “Pag. 2, line 56”.
- Line 55: more details on the nature of amphotericin B toxicities would be welcome
We have added some more information about the toxicity of amphotericin B “Pag 2, lines 57 to 60”.
- Line 56: consider rewording this last sentence, as “problems” is a rather vague term
More information was added about the problems caused by the azoles “Pag 2., lines 60 to 63”.
- Line 58: the term "derivatives" may be preferable to "derivates"; at the discretion of the authors
The suggestion was accepted “Pag. 2, line 65”.
- Line 59: nitrofuran derivatives such as nifurtimox are also used in human medicine; this should be mentioned more evidently
Information on the use of a nitrofuran derivative in human medicine was mentioned “Pag. 2, lines 66 and 67”.
- Line 60: same remark concerning the “toxicity-related” issues of nitrofurans that should be more clearly stated
Modifications were made on the toxicity of nitrofurans “Pag 2, lines 67 to 70”.
- Line 67: please remove the coma before “such as cancer”
The comma has been removed “Pag. 2, line 77”.
- Line 70: please remove “models”
The word has been removed “Pag 2, Line, 80”.
- Line 73: the bibliographic reference corresponding to the articles describing the tested compounds should be mentioned here, in addition to being included in the experimental part
References describing the compounds have been added “Pag. 2, Line 83”.
- Line 92: please write “the in vivo environment”
The term was written “Pag. 5, lines 102 and 103”.
- Line 113: please write “subcultured”
The word has been corrected “Pag 5, line 123”.
- Lines 111, 115 and 116: please replace the coma by a dot in “0,1%”, 0,1%” and “0,6%”, respectively
The comma has been removed “Pag. 5 lines 122 and 126”.
- Line 116: please write “0.008% L-cysteine” without a space between the number and the percent symbol, and without a capital letter at cysteine.
The remarks have been corrected “Pag 5, line 127”.
- Line 125: the term "derivatives" may be preferable to "derivates"
The suggestion was accepted “Pag 6, line 137”.
- Line 137: please remove “the” before “Trypan blue”
The word has been removed “Pag 6, line 147”.
- Line 138: please replace the “x” in “1 x 106” by a proper multiplication symbol, as it is the case right after
The remarks have been corrected “Pag 6, line 149”.
- Line 171: please reconsider rewording this sentence as it is not grammatically correct
The sentence has been rewritten “Pag 7, line 179 and 180”.
- Lines 257 and 260: please consider rephrasing these sentences so that they do not begin with a number or consider writing these numbers in words.
Both sentences have been rewritten “Pag. 9, lines 264 and 265; 267 and 268”.
- Line 300: please write “2 h” and not “2 hours”
The remark has been corrected “Pag 10, line 306”.
- Line 301: please check the possible extra space before “1x PBS”
The extra space has been removed “Pag 10, line 307”.
- Line 314: for all the results given in µg/mL, wouldn’t it be more relevant to express them in molar concentration to overcome the variation in molecular weight between each molecule? This would sound more rigorous
We agree that the expression of results in molar would be much more rigorous. However, the conversation to molar would change the whole article and moreover, most of the works that are used as a unit comparison are in ug/ml. But we thank the reviewer for suggestion and will start to consider the change to molar in the next papers, to make it more suitable.
- Line 343-343: please add space before and after the “<” symbol, as it is done afterwards
- Line 372-373: same remark, please add space before and after the “<” symbol
- Line 376: same remark, please add space before and after the “<” symbol
In all indicated lines, spacing has been added “Pag. 13, lines 351 and 352; lines 362 and 363, line 366”.
- Table 3: please replace all the “x” by a proper multiplication symbol
The multiplication symbol has been added throughout Table 3 “Pag. 13”
- Line 395: please add a capital letter at “Viability”
The title has been corrected “Pag. 14, line 376”.
- Line 442: why the cytotoxicity results are given in inhibitory concentrations (IC50) and not cytotoxic concentrations (CC50) if it is the cytotoxicity of these compounds that is measured? This point needs to be clarified.
Both terminologies refer to the same thing, a concentration capable of inhibiting 50% of the cells, but we agree that, as it is about cytotoxicity, the most conventional and usual is CC50. We appreciate your correction. All IC50 terminology in the text has been modified by CC50. “Pag. 8, line 251; Pag. 10, line 315, Pag. 17, line 416, Pag. 17 – Table 4; Pag. 18, line 426, Pag. 19, Table 5, Pag. 19, line 431, Pag. 24, line 543”.
- Tables 4 and 5: as several cytotoxicity values are “greater than” a given value, the resulting selectivity indices are also “greater than” the ratio calculated. Please add the “>” symbol before the SI values concerned (6 occurrences in Table 4; 8 occurrences in Table 5)
In tables 4 and 5, the symbol “>” was added to the values corresponding to the SI “Pag 17, 18 and 19 – Table 4 and 5”.
- Line 480: please remove “The” before “Compound 11”
- Line 481: please remove “the” before “compound 7”
- Line 483: please remove “the” before “compound 32”
All “the” have been removed. “Pag. 20, lines 453, 454 and 455”.
- Line 506, rewording suggestion: “Although effective, therapeutic options for treating histoplasmosis are limited and with high risks of toxicity for the patient. Therefore, …”
The suggestion was accepted, and the sentence modified “Pag. 23, line 479 and 480”.
- Line 516: please remove “the” before “compounds 1 and 17”
The word has been removed “Pag. 23, line 489”.
- Line 526-527: please reconsider rewording as this sentence, although understandable, is quite unclear
The sentence has been rewritten “Pag 23, line 498 and 499”.
- Line 555: please write “the 3D cell culture”
The term was written “Pag. 23, line 511”.
- Line 572: please consider replacing “IC50” by “CC50”
All IC50 terminology in the text has been modified by CC50 “Pag. 8, line 251; Pag. 10, line 315, Pag. 17, line 416, Pag. 17 – Table 4; Pag. 18, line 426, Pag. 19, Table 5, Pag. 19, line 431, Pag. 24, line 543”.
- Line 577: a comparison to reference compounds would be expected in the discussion
We have added a simple discussion of reference drugs “Pag 24, lines 549 to 555”.
- Discussion part: one of the well-known toxicities of nitro-containing molecules such as nifurtimox is their mutagenicity and genotoxicity. The corresponding in vitro evaluations (Ames test and comet assay) on compounds 7 and 11 would be of great value in this study. If these tests can’t be undertaken as a part of this work, this topic should still be discussed in depth.
Thanks for the suggestion, a discussion was held on the toxicity of nitrofurans “Pag 24 and 25, lines 556 to 563”.
- Line 625-626: this statement is very premature (as no extensive toxicity, physicochemical, stability and especially in vivo studies were undertaken on these molecules) and should be reworded. Suggestion: … derivatives are promising molecular scaffolds for the development of original anti-histoplasmosis drug candidates”
The suggestion was accepted, and the sentence was modified “Pag. 26, line 611 and 612”.
- Line 652: bibliographical references should be reworked in depth to be all formulated in a homogeneous way in accordance with the citation style of the Multidisciplinary Digital Publishing Institute. See the publisher's website.
The authors thank the reviewer for all suggestions. Regarding references, the citation form was made by the EndNote application, and as indicated I consulted the publisher's website and installed the MDPI citation form.

Round 2
Reviewer 1 Report
I satisfy the authors’ revision in response to my comments. The new version was improved after adding some necessary information and references.
Reviewer 3 Report
I acknowledge the changes made by the authors in their manuscript.